# Three-Dimensional Cell Co-Culture Liver Models and Their Applications in Pharmaceutical Research

**DOI:** 10.3390/ijms24076248

**Published:** 2023-03-26

**Authors:** Yinping Ma, Lei Hu, Jianhua Tang, Weiwei Guo, Yujie Feng, Yanmiao Liu, Fushan Tang

**Affiliations:** 1Department of Clinical Pharmacy, Key Laboratory of Basic Pharmacology of Guizhou Province and School of Pharmacy, Zunyi Medical University, Zunyi 563006, China; 2Key Laboratory of Basic Pharmacology of Ministry of Education and Joint International Research Laboratory of Ethnomedicine of Ministry of Education, Zunyi Medical University, Zunyi 563006, China; 3Key Laboratory of Clinical Pharmacy of Zunyi City, Zunyi Medical University, Zunyi 563006, China; 4Cancer Research UK Manchester Institute, The University of Manchester, Cheshire SK10 4TG, UK; 5School of Preclinical Medicine, Zunyi Medical University, Zunyi 563006, China

**Keywords:** liver, 3D cell co-culture, HepaRG, liver sinusoidal endothelial cells, in vitro models, pharmaceutical research

## Abstract

As the primary site for the biotransformation of drugs, the liver is the most focused on organ type in pharmaceutical research. However, despite being widely used in pharmaceutical research, animal models have inherent species differences, while two-dimensional (2D) liver cell monocultures or co-cultures and three-dimensional (3D) liver cell monoculture in vitro liver models do not sufficiently represent the complexity of the human liver’s structure and function, making the evaluation results from these tools less reliable. Therefore, there is a pressing need to develop more representative in vitro liver models for pharmaceutical research. Fortunately, an exciting new development in recent years has been the emergence of 3D liver cell co-culture models. These models hold great promise as in vitro pharmaceutical research tools, because they can reproduce liver structure and function more practically. This review begins by explaining the structure and main cell composition of the liver, before introducing the potential advantages of 3D cell co-culture liver models for pharmaceutical research. We also discuss the main sources of hepatocytes and the 3D cell co-culture methods used in constructing these models. In addition, we explore the applications of 3D cell co-culture liver models with different functional states and suggest prospects for their further development.

## 1. Introduction

The liver is the body’s largest digestive gland, responsible for more than 500 functions, including metabolism and detoxification [1,2]. It is mainly composed of hepatocytes and non-parenchymal cells, such as liver sinusoidal endothelial cells (LSECs), hepatic stellate cells (HSCs), and Kupffer cells (KCs), which work together to regulate liver function through direct or indirect cell crosstalk events [3,4]. 

As the primary organ of drug exposure, the liver plays a critical role in drug metabolism and toxicity, necessitating the development of liver models that can more accurately predict the efficacy and toxicity of drugs in humans for drug discovery [5,6,7,8,9]. In vitro liver models, including 2D and 3D liver cell monoculture and co-culture models, offer advantages over animal models, in terms of cost, ethical considerations, and efficiency. However, 2D liver cell models lack physiological relevance, while 3D liver cell monoculture models fail to represent the complex interactions between parenchymal and non-parenchymal cells [10,11,12]. Therefore, the development of 3D co-culture liver models that mimic the natural liver tissue structure and cell interactions offers the potential for more accurate in vitro prediction of drug response sensitivity in humans. 

The previous frequently used prediction tools tended to fail to reliably predict drug efficacy and toxicity in humans, leading to high attrition rates in drug development [13]. For example, troglitazone, trovafloxacin, and nefazodone have been withdrawn from the market due to their severe hepatotoxicity [14]. Three-dimensional cell co-culture liver models can provide a greater predictability of drug response in humans and have been applied in many studies to predict the in vivo metabolism and clearance of candidate or marketed drugs [15,16]. For example, one study showed that Diclofenac exhibited phase I and phase II metabolic profiles and toxic effects similar to those in vivo in a microfluidic liver chip model [17]. A 3D cell co-culture liver model based on 3D bioprinting technology and microfluidic technology was applied to evaluate the efficacy of Metuzumab and showed similar results to those obtained in animal experiments and clinical trials of anti-cancer drugs with similar antitumor mechanisms [18]. Spheroid cultures of Primary human hepatocytes (PHHs), LSECs, KCs, and HSCs were used as models of nonalcoholic fatty liver disease (NASH), to evaluate the efficacy of the drug candidates Selonsertib and Firsocostat against NASH, with the results correlating with the effects in patients participating in clinical trials [19].

In this review, we first analyze the potential advantages of 3D cell co-culture liver models for pharmaceutical research based on the discussion of the unique structure and function of primary parenchymal and non-parenchymal cells in the liver; then, we introduce the main sources of hepatocytes in 3D cell co-culture liver models and the commonly used methods for the construction of 3D cell co-culture liver models; finally, we discuss 3D cell co-culture liver systems with different functions and their applications. 

## 2. The Main Cellular Components of the Liver and Their Functions

Hepatic lobules with a hexagonal structure constitute the basic structural and functional unit of the liver (Figure 1). A central vein runs through the center of each lobule and is responsible for transporting intrahepatic metabolites. Surrounding the lobule is the portal triad, consisting of the hepatic portal vein, hepatic artery, and bile duct. The hepatic portal vein carries nutrient-rich blood to the liver, while the hepatic artery supplies oxygen to the liver [20]. 

Hepatocytes are the most abundant cell type in the liver, and they form hepatocyte plates that radiate out from the central vein. The adjacent liver plates are separated by hepatic sinusoids, which are capillary-like structures formed by the confluence of the terminal branches of the hepatic portal vein and the hepatic artery. In addition, adjacent hepatocytes are connected by the gaps that form bile canaliculi, which collect and transport bile. Liver sinusoidal endothelial cells (LSECs) form the liver sinusoidal wall and facilitate material exchange between the hepatocytes and the sinusoidal blood in the Disse space, which is the space between the hepatocytes and the LSECs [21]. Kupffer cells (KCs) are located in the hepatic sinusoid lumen and extend along the LSECs. Hepatic stellate cells (HSCs) are located in the space of Disse and, together with the space of Disse components, separate the hepatocytes from the LSECs [21]. It should be emphasized that the structure and function of hepatic lobules are regulated by the intricate interactions between the various liver cells (Table 1). 

## 3. Potential Advantages of 3D Cell Co-Culture Liver Models for Drug Evaluation

It is necessary to further integrate the structure and function of the liver in vivo into liver models in vitro, as much as possible, to improve the reproducibility of in vitro experimental results in humans. In vitro liver models have evolved from 2D culture to the sandwich culture and then to the 3D culture of liver cells. Recently, the emergence of multicellular 3D cell co-culture models has revolutionized the construction of in vitro liver models. Although traditional 2D liver cell culture is inexpensive and easy to perform, the liver cells grow in a flat state under these conditions, resulting in cytoskeletal changes and loss of cell polarity within 2–3 days [33]. Most critically, drug metabolic enzymes, which play a key role in hepatic drug metabolism, have a low expression and activity under 2D conditions, which determines the ineffectiveness of 2D liver cell monoculture and co-culture models for research on the prediction of hepatic drug metabolism, drug interactions, and toxic reactions mediated by drug metabolic enzymes. Fortunately, the transformation from a 2D culture to a 3D culture mode for liver cells effectively overcomes the environmental homogeneity limitations of 2D systems [34,35]. Three-dimensional systems can be self-aggregated cell spheres, extracellular matrix (ECM)-like 3D network scaffolds made of natural or synthetic materials, and organ-like systems constructed using 3D technology, which have oxygen, nutrient, and various biological factor gradients and 3D tissue structures, where cell–cell and cell–ECM interactions can occur. This culture mode, which is closer to the complex microenvironment in vivo, largely contributes to the improved functional activity of liver cells in vitro [36]. 

However, it has recently been realized that 3D liver cell monoculture models ignore the direct or indirect signal exchange of heterotypic cells in liver tissue. Thus, 3D co-culture liver models that integrate hepatic heterotypic cell interactions and 3D microenvironmental signaling stand out, with many advantages. Several studies have demonstrated that 3D cell co-culture liver models exhibit better predictive ability than 2D and 3D monocultures and 2D co-cultures in drug-induced hepatotoxicity, hepatic drug metabolism, and drug interaction studies, since cell growing environment and signaling regulation between heterotypic cells in 3D cell co-culture allow them to exhibit more relevant liver environments and phenotypes in vivo, such as reproducible liver sinusoidal structures, long-term maintenance of liver cell activity and function, and increased expression of genes related to liver-specific functions [37,38,39]. The 3D co-culture model of hepatoma cells and fibroblasts constructed by Al Hrout et al. [40] exhibited an increased expression of genes and proteins associated with hepatocellular carcinoma development and a poor prognosis compared to 2D or 3D monocultures of hepatoma cells, and it more closely resembled the developmental characteristics of hepatocellular carcinoma in vivo and facilitated a more accurate assessment of the anticancer effects of anticancer drugs in vitro. In summary, 3D cell co-culture liver models can effectively improve the functional stability of hepatocytes in vitro and ensure the reliability of in vitro drug prediction results and thus are expected to provide a reliable new platform for pharmaceutical research.

## 4. Hepatocyte Sources of 3D Cell Co-Culture Liver Models

Hepatocytes are the predominant cell type in the liver and are responsible for most of its functions, making them the most important cell type in 3D cell co-culture liver models in vitro. Their value in pharmaceutical research lies in their ability to express specific liver functions, such as protein and urea synthesis, as well as drug metabolic enzyme expression levels, particularly CYP450 enzymes. Additionally, their sensitivity to stimuli such as toxins or drugs is critical for evaluating drug efficacy and toxicity. 

### 4.1. Primary Hepatocytes

Primary hepatocytes (PHs) are widely considered the gold standard for constructing in vitro liver models, with both animal and human sources being utilized [41]. Primary human hepatocytes (PHHs), in particular, are preferred by researchers in drug interaction, drug metabolism, and drug toxicity studies, due to their secretion properties and metabolic enzyme activities being comparable to those of in vivo hepatocytes, and they can reliably reflect in vivo liver properties [42]. However, the stable expression of their functional properties is highly dependent on the signal regulation of the microenvironment, which means that PHHs will quickly experience phenotypic changes, decreased cell activity, and loss of cell membrane polarity under a single 2D environment [43]. 

The in vitro culture conditions of PHHs have been improved, to enhance their functional activity, with studies such as Hu et al. [44] and Mazzocchi et al. [45] utilizing a 3D culture environment composed of Matrigel and a 3D bio-printed model, respectively. These studies have shown that PHHs can maintain their morphology and function for several months, with improved sensitivity to hepatotoxic agents. However, applications have been greatly restricted by the limited number of donors, ethical constraints, and genetic variability among donors [46]. Animal PHs are more readily available than PHHs, but due to the species differences between animals and humans, their applications are also limited by poor reproducibility. 

### 4.2. Hepatoma Cell Lines

HepaRG, HepG2, Huh7, and Hep3B are common hepatocellular carcinoma-derived cell lines used for constructing in vitro liver models because they are readily available, easy to culture, highly proliferative, and phenotypically stable, with less inter-donor variability compared to PHHs [47]. However, the high proliferative potential of these cell lines is usually accompanied by a hypodifferentiated phenotype, which leads to a lack of specific liver functions, especially in the level of enzyme expression (especially the phase I and II metabolic enzymes involved in drug metabolism), protein and urea synthesis, and a much lower sensitivity to toxic reactions than PHHs [48]. For example, HepG2, Huh7, and Hep3B only express human liver-specific CYP450 enzymes at low levels, about 5–15% of that of PHHs, which may limit their predictive ability for hepatotoxicity caused by parent drug metabolites. Moreover, although they have similar expression levels of most phase II metabolic enzymes to PHHs [49,50], their reliability as a model for drug-induced hepatotoxicity assessment should be further evaluated. 

In contrast, HepaRG cells, which were identified from a patient with hepatocellular carcinoma associated with hepatitis C virus infection, can differentiate into hepatocyte-like and bile duct-like cells after high-density exposure to dimethyl sulfoxide (DMSO) [51,52]. They acquire a highly differentiated phenotype and typical expression of liver function, including CYP450 enzyme activity (except CYP2D6) and albumin and urea synthesis similar to PHHs. They also express nuclear receptors such as the pregnane X receptor and the homeodomain androstane receptor [46]. Furthermore, they can form bile duct structures and express hepatic sinusoidal and bile canalicular transporters that are not found in other hepatoma cell lines, which play an important role in endogenous and exogenous material transport [53,54]. These structures and signals make HepaRG cells the most promising alternative to PHHs for in vitro drug evaluation. 

However, some studies have shown that HepaRG cell-derived hepatocyte-like cells (HepaRG-tdHep), when stimulated by inflammatory factors TNF-α, IL-6, and TGF-β, can retro-differentiate into progenitor cells with stem cell characteristics, accompanied by the decrease in expression of liver-specific functions. In particular, TNF-α can indirectly induce expression deficiency of liver-specific markers (ALDOB and CDH1) in HepaRG-tdHep through activation of the NFkB gene network, while IL-6 can directly reduce the expression levels of ALDOB and CDH1. Additionally, TGF-β can induce the epithelial-to-mesenchymal transition (EMT) of HepaRG-tdHep, to regulate their retro-differentiation process. Interestingly, the retro-differentiation of HepaRG-tdHep induced by TNF-α, IL-6, and TGF-β is not reversed by DMSO [55,56]. Furthermore, the expression level of the liver-specific functions in HepaRG cell lines can also be regulated by their seeding density, since HepaRG cells differentiated into hepatocyte-like cells can retro-differentiate into hepatic progenitor cells through low-density seeding [57]. Therefore, the phenomenon of retro-differentiation exhibited by HepaRG cells under the influence of multiple factors is a point to be noted during their application.

### 4.3. Stem Cells

Human induced pluripotent stem cells (HiPSCs), embryonic stem cells (ESCs), and adult stem cells (ASCs), such as mesenchymal stem cells, hepatic stem cells, and hematopoietic stem cells, have demonstrated strong self-replication and self-renewal capabilities in vitro. HiPSCs and ESCs, in particular, have the potential to differentiate into almost any cell type in the human body, under certain conditions. Moreover, HiPSCs can be obtained from patient biopsies for personalized research [58,59]. In the context of the global scarcity of liver donors, stem cells capable of differentiating into hepatocyte-like cells under specific conditions can serve as potential resources for liver disease treatment, regeneration, and in vitro modeling. 

However, it can be difficult for stem cells to differentiate into functionally mature hepatocytes in vitro, and there can be challenges such as immune rejection and resource shortage [60]. Although HiPSCs can effectively avoid ethical, immunogenicity, and resource shortage problems, their ability to differentiate directionally into mature hepatocytes is limited, and their derived hepatocytes exhibit more of the phenotype and characteristics of fetal hepatocytes, such as expressing high levels of alpha-fetoprotein and lower mature liver-specific metabolic functions [59,61,62,63]. To improve the ability of stem cells to differentiate into mature hepatocytes in vitro, it is essential to optimize their in vitro differentiation conditions. For example, the CYP450 enzyme activity and protein expression in HiPSC-derived hepatocytes increased when cultured in a 3D environment consisting of cellulose nanofibril (CNF) gel [64]. However, despite the improvement in culture conditions, their degree of differentiation is still unsatisfactory, which is a crucial factor limiting their application. Therefore, it is important to further explore methods for improving the stable differentiation of stem cells into hepatocytes, making them a reliable source of hepatocytes for research and potential clinical applications.

## 5. Common Co-Culture Methods for 3D Cell Co-Culture Liver Models

The development of liver function is known to be closely related to the environment–cell complex and heterotypic or homotypic cell–cell communication, which includes both direct physical contact communication and indirect paracrine signaling exchange between cells. To more realistically reproduce the intercellular information transfer in the liver, 3D cell co-culture liver models can be mainly divided into direct 3D co-culture and indirect 3D co-culture, based on their cell culture conditions. These can be realized with various 3D techniques, such as self-aggregating multicellular spheroids, 3D liver organoids, and 3D scaffold co-culture of liver cells for achieving direct 3D co-culture, while indirect 3D co-culture can be achieved based on Transwell chambers. Furthermore, 3D bioprinting co-culture of liver cells and microfluidic multicellular liver chips can be either direct 3D cell co-culture liver models or indirect 3D cell co-culture liver models (Table 2).

### 5.1. Direct 3D Co-Culture

Direct 3D co-culture involves mixing two or more different types of liver cells into self-assembling spheroids or culturing them together in a 3D environment that simulates the structure of native liver tissue (Figure 2). This 3D environment can typically be constructed with natural or synthetic materials that are low-toxicity or non-toxic, biocompatible and biodegradable, such as collagen, fibrin, alginate, and hydrogel. Pingitore et al. [69] generated mixed cell spheroids of HepG2 cells and immortalized hepatic stellate cells (LX-2) at a 1:1 or a physiological ratio of 24:1 in ultra-low attachment 96-well plates, to construct a non-alcoholic fatty liver-like organ system. In addition, Ahmed et al. [86] sequentially seeded primary human LSECs, HSCs, and hepatocytes on hollow fiber membranes (HF), to mimic the contact-layered structure between heterotypic cells in liver tissue. 

Direct 3D co-culture allows for close contact between different liver cells, which enables signal communication between them through direct cell–cell adhesion, paracrine secretion of soluble cytokines, cell–ECM adhesion, and other mechanisms. In a multicellular organ such as the liver, direct contact between heterotypic cells plays an important role in full liver function. However, it is important to consider the actual distribution of co-cultured cells in the liver tissue in vivo when deciding on the rationality of direct co-culture. Direct contact co-culture in vitro of cell types that do not normally have direct contact in vivo may be unnecessary. Furthermore, tight junctions between heterotypic cells also affect the difficulty of cell separation, which can be disadvantageous for experiments requiring a single-cell type analysis. Therefore, the decision to use direct co-culture should be based on the type of co-culture cells and the purpose of the experiment.

### 5.2. Indirect 3D Co-Culture

Indirect 3D co-culture involves the culture of two or more types of liver cells in a 3D environment with a physical separation system (Figure 3), which does not allow direct contact between the cells present on either side of the physical separation system, and signals between them are communicated through soluble cytokines. Common materials for physical separation systems are Transwell cell culture plates and various natural or synthetic materials. Otsuka et al. [87] constructed a 3D layered co-culture system from rat PHs and NIH/3T3 fibroblasts using Transwell cell culture plates as a physical separation system and explored the interaction between heterotypic cells. Kang et al. [23] seeded rat PHs and LSECs on both sides of a 6-well Transwell membrane precoated with rat tail collagen (LSECs on the top and hepatocytes on the bottom). They achieved a 3D layered co-culture of hepatocytes and LSECs, in which the 6-well Transwell membrane precoated with rat tail collagen mimicked the Disse spatial structure. The morphology of hepatocytes remained normal, and the expression of CYP450 enzymes remained similar during 39 days of culture, demonstrating the importance of soluble cytokines from LSECs for maintaining hepatocyte function. In addition, some researchers have also achieved indirect 3D co-culture of HepG2 and NIH/3T3 fibroblasts with the application of 3D bioprinting technology. They demonstrated that co-culture of HepG2 and NIH/3T3 fibroblasts exhibited a better hepatocyte function compared to monocultures of HepG2 [88]. In conclusion, indirect co-culture systems can effectively avoid unnecessary cell contact, facilitate the separation of co-cultured heterotypic cells, and may become promising tools for studying non-contact liver cell communication in vivo.

## 6. Types of 3D Cell Co-Culture Liver Model

The liver microenvironment is complex and dynamic, whereby the hepatocytes are surrounded by the ECM, which provides structural support and signaling molecules essential for their growth, proliferation, adhesion, and differentiation. Furthermore, hepatocytes communicate with non-parenchymal cells, either through direct cell–cell contact or indirect signaling mechanisms, leading to distinct functional states of the liver. Hence, different 3D co-culture combinations of hepatocyte and non-parenchymal cells can reproduce liver activity in diverse functional states in vivo, for different research purposes. This section outlines the 3D cell co-culture liver systems representing various functional states.

### 6.1. Three-Dimensional Cell Co-Culture Models for Enhanced Hepatocyte Function

In liver tissue, hepatocytes are surrounded by the hepatic sinusoidal network formed by LSECs, which is crucial for maintaining hepatocyte activity and functional homeostasis, as the basis of material exchange in hepatocytes [89]. Co-culture of LSECs and hepatocytes in vitro has been shown to be able to achieve a vascularized 3D liver tissue structure, where the presence of the vascular-like tubular network provides nutrients and oxygen to support the function of hepatocytes in vitro and maintain the elimination of metabolic waste products [90]. Additionally, LSECs can induce upregulation of vascular secretory factor Wnt2 and hepatocyte growth factor (HGF) through the Id1 pathway, both of which are essential for hepatocyte proliferation [89]. 

Studies have evaluated the effects of co-culturing LSECs with hepatocytes in 3D environments using various combinations of cells. Nahmias et al. [91] constructed monocultures of rat PHs and co-cultures with rat LSECs, human umbilical vein endothelial cells (HUVECs), or 3T3-J2 fibroblasts in a 3D environment of collagen gels, respectively. The ability of hepatocytes to clear low-density lipoprotein (LDL) in these cultures was assessed. The results showed that the 3D co-culture of PHs-LSECs not only reproduced the liver sinusoidal structure but also improved the ability of hepatocytes to clear LDL, which was not observed in other co-cultures. Ardalani H et al. [92] constructed mixed cell spheroids, including HiPSCs-derived hepatocytes (iHEP) and LSECs. The iHEP/LSECs spheroids showed a higher albumin and urea secretion capacity, as well as CYP3A4 and CYP2C9 activities, in the presence of DMSO, compared with other spheroids of iHEP, iHEP/HSCs, iHEP/cholangiocyte (CCs), iHEP/iPS-derived endothelial cells (iECs), or iHEP/LSECs/HSCs. Moreover, the presence of bile duct structures and capillary-like endothelial networks in iHEP/LSECs spheroids was confirmed through immunostaining, indicating that the 3D spheroid of LSECs and iHEP promoted the in vitro maturation of iHEP. German et al. [93] cultured HepaRG cells alone or in combination with LSECs or HUVECs in a 2D environment or a porous 3D environment composed of chitosan–gelatin. The results showed that the 3D co-culture of HepaRG/LSECs showed the highest HepaRG cell viability and APAP transformation capacity, whereas a higher CYP3A4 activity was exhibited in 3D co-cultures of HepaRG/HUVECs. 

In summary, co-culturing LSECs with hepatocytes in 3D environments can recapitulate key structures of the liver sinusoids in vivo, which is important for research on enhancing in vitro hepatocyte function.

### 6.2. Three-Dimensional Cell Co-Culture Models of Liver Fibrosis

Hepatic fibrosis is a liver disease characterized by the accumulation of excess extracellular matrix proteins in the liver. The induction process is usually co-regulated by multiple types of liver cells. However, it is noteworthy that the activation of HSCs and their transformation to a myofibroblast-like cell phenotype are followed by excessive production of extracellular matrix proteins, which is the main mechanism by which hepatic fibrosis occurs. This dictates that the main target of hepatic fibrosis research is HSCs [94]. 

On this basis, a series of in vivo-relevant 3D cell co-culture liver fiber models have been developed in recent years. For instance, Prestigiacomo et al. [95] mixed rat PHs, HSCs, and KCs into 3D spheroid cultures using the hanging drop technique and explored the response of 3D spheroid cultures of PHs/HSC/KC and PHs with the stimulation of the pro-fibrotic factors TGF-β1 and LPS. When treated with TGF-β1 (1 ng/mL) or LPS, increased expression of α-SMA, an activating HSC-related gene, or an inflammatory response triggered by cytokines was detected in PHs/HSC/KC spheroid cultures. At the same time, a similar phenomenon was not observed in PHs spheroids. To construct a more representative in vitro model of liver fibrosis, Mannaerts et al. [96] prepared mixed cell spheroids of mouse PHs and HSCs in a 1:2 cell ratio. They exposed 2D cultures of HSCs and mixed spheroids of PHs and HSCs to APAP, to induce liver injury, to assess the activation of HSCs in a state of liver injury. It was found that mixed spheroids of PHs/HSCs better reproduced liver injury-dependent HSC activation, exhibiting similar gene upregulation of activated HSCs in vivo, such as Bicc1, Mfap4, Fmod, Oas1g, and Col6a3, whereas 2D cultures of HSCs showed low expression of these genes after the same treatment. Similarly, Leite et al. [97] developed a spheroid model of HepaRG/HSC. Their data showed that 3D spheroids of HepaRG/HSCs maintained functional activity for 21 days and allowed the drug to activate HSCs in a hepatocyte-dependent manner, as manifested by pro-fibrotic compound stimulation with massive collagen secretion and deposition. Existing studies support these experimental results; when liver injury occurs, hepatocytes can alter their gene expression and secretion profile, resulting in high levels of expression of the Notch protein, osteopontin, TGFβ, NADPH oxidase, and NLRP3 inflammasome proteins involved in HSC activation, which can further induce fibrosis [98,99,100]. Likewise, in a Transwell culture system, Yu et al. [101] co-cultured free fatty acid-induced senescent hepatocytes in a layer with primary HSCs. They demonstrated HSC activation and liver fibrosis associated with hepatocyte senescence. Studies have shown that the aging of hepatocytes can lead to a decline or loss of activity of the Nrf2-ARE antioxidant pathway, resulting in oxidative stress, which activates HSCs and induces fibrosis. 

However, apart from hepatocytes, LSECs and KCs also play a vital role in HSC activation. Pathologically, LSECs undergo rapid loss of their fenestrated structure, transforming into a capillary phenotype, which is often accompanied by the production of factors that promote HSC activation, such as fibronectin (FN), sphingosine kinase 1 (SphK1), and adipocyte fatty acid-binding protein (A-FABP) [102,103,104]. Furthermore, since liver fibrosis is typically accompanied by a hepatic inflammatory response, KCs, as liver resident macrophages, are the primary players in this response and can induce HSC activation and the liver fibrosis process once activated. One study on a CCl4-induced mouse model of liver fibrosis highlighted significant upregulation of the KC-derived TIM-4, a class of pro-fibrotic factors [105]. In another study, it was confirmed that a damage-related factor, HMGB1, was derived from KCs during liver injury and could stimulate HSCs to secrete a large amount of type I collagen through the RAGE signaling pathway, participating in the liver fibrosis process [106]. Under oxidative stress, KCs can also generate reactive oxygen species (ROS), which activate HSCs and induce liver fibrosis [107]. 

In summary, liver fibrosis in vivo is regulated by various liver cells, including HSCs, hepatocytes, LSECs, and KCs. However, there are currently few co-culture models of liver fibrosis, and only a few studies have reported the co-culture of LSECs with HSCs, hepatocytes, and/or KCs, to develop in vitro models of liver fibrosis. Therefore, future studies must integrate different types of liver cells into a 3D environment for co-culture, to construct a more representative in vitro model of liver fibrosis.

### 6.3. Three-Dimensional Cell Co-Culture Model of Drug-Induced Liver Injury

During drug discovery, preclinical experimental models often fail to predict drug-induced hepatotoxicity, resulting in clinical trial failures and significant economic losses. Hepatocytes are typically used as the gold standard for drug screening models, due to their unique functional activities. However, the role of KCs as liver resident macrophages cannot be ignored, especially in immune-mediated drug-induced liver injury, as they perform potent scavenger and phagocytosis functions. Once the immune environment is dysregulated, they can directly contact hepatocytes or activate into the M1 phenotype and produce various inflammatory cytokines, growth factors, and ROS, leading to liver injury [108]. The homogeneity of hepatocyte monoculture models is one of the reasons for the low reliability of hepatotoxicity assay data. To address this issue, some investigators co-cultured iPSC-derived KCs (iKCs) with cryopreserved PHHs in a 3D system, which demonstrated improved sensitivity for predicting acetaminophen- and trovafloxacin-induced hepatotoxicity and chlorpromazine-induced cholestasis compared with 3D monocultures of PHH [109]. 

In short, KCs play a vital role in the hepatotoxicity response, and 3D co-culture systems containing hepatocytes and KCs have the potential to provide more accurate and efficient tools for preclinical drug-induced hepatotoxicity assessment in the future. However, more research is needed to confirm the functional properties of these co-culture systems for general applications in toxic drug screening.

## 7. Conclusions and Application Prospects of 3D Cell Co-Culture Models

In conclusion, 3D co-culture of heterotypic liver cells is a promising tool for preclinical drug screening, due to its ability to recapitulate the complex cellular communication and liver phenotype, functioning closer to in vivo compared with 2D monoculture or 2D co-culture of heterotypic liver cells. For instance, 3D liver co-culture systems containing HSCs and hepatocytes have demonstrated their potential in reproducing liver fibrosis onset and progression in vitro. Similarly, 3D co-culture systems comprising hepatocytes, and KCs may be a better choice to assess drug-induced liver injury, because most drug-induced hepatotoxicity is immune-mediated. In addition, the presence of more physiologically relevant hepatocyte functional activities in 3D co-culture models based on hepatocytes and LSECs could provide realistic in vitro experimental results to generalize hepatic drug interactions, drug metabolism, and enzyme induction in vivo. Finally, suitable liver cell co-culture models can be selectively constructed according to the purpose of the study, given the varying levels of function of hepatocytes induced by different hepatic non-parenchymal cells. 

Furthermore, to improve the reproducibility of in vitro evaluation data for drugs in humans, future research will focus on developing in vitro liver models that mimic the complex cell crosstalk and cell growth microenvironment present in vivo. However, hepatic non-parenchymal cells, such as LSECs and HSCs, are currently less well-studied than hepatic parenchymal cells (PHHs or hepatocellular carcinoma cell lines). Unfortunately, in the absence of suitable in vitro culture conditions, they lose their characteristic phenotypes and functions after a short period of in vitro culture. Therefore, developing optimized in vitro culture conditions for hepatic non-parenchymal cells, to improve their in vitro expansion capacity and availability, is a major challenge and one that urgently needs to be overcome in future studies. Meanwhile, selecting optimized media components that simultaneously maintain the activity and functional stability of different co-cultured cell types is essential. Notably, developing in vitro cell culture systems with more liver ECM characteristics, to simulate the complex microenvironment and cellular signaling in vivo, may be necessary. Liver ECM components, such as collagen, fibronectin, laminin, hyaluronic acid, and some growth factors, play a pivotal role in cell adhesion, growth, migration, and differentiation processes, through various signaling modulations. 

Moreover, the emergence of microfluidics in recent years has provided a new opportunity to improve cell culture conditions in vitro. Combining microfluidics with 3D cell culture technology could potentially overcome the limitations of traditional static 3D cell culture and allow a sustained blood perfusion physiological microenvironment similar to that in vivo for cell culture in vitro. This approach is promising for providing more sensitive nutrient and oxygen gradients, as well as metabolic waste excretion for cells. In particular, microfluidic liver chips, and even microfluidic multi-organ chips, with their more precise cellular and environmental regulation, are expected to improve the life activity and function of cells in vitro from several aspects, providing more systematic and realistic in vitro assessment data of the real response of drugs in humans, for drug development and post-marketing re-evaluation. However, the technical complexity and high cost of microfluidic chips are challenges that must be considered and overcome. 

In conclusion, 3D cell co-culture liver models with more complex cellular communication and microenvironmental characteristics will become a powerful tool in future drug development and screening studies, improving the reliability of drug evaluation data in vitro. Addressing the challenges of in vitro culture conditions for liver cells and developing further liver ECM-like cell culture systems and microfluidic chips are essential steps in this direction.

## Figures and Tables

**Figure 1 ijms-24-06248-f001:**
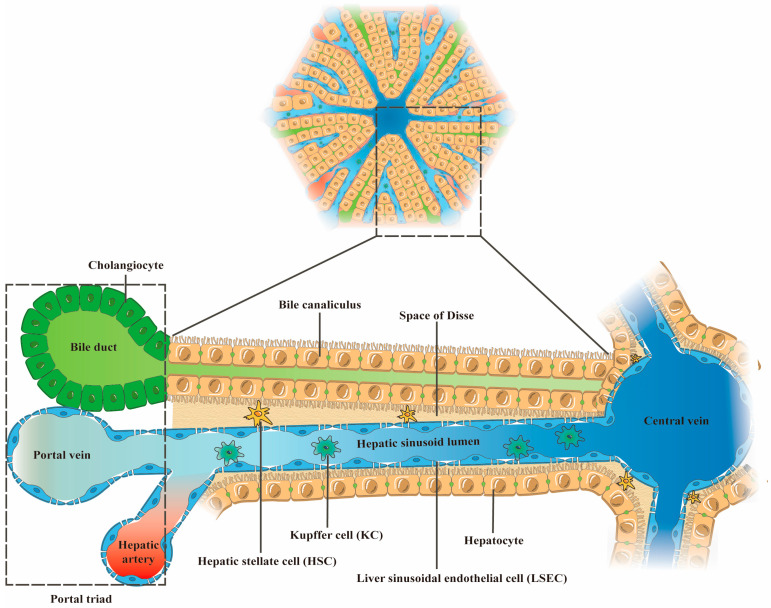
Schematic diagram of liver lobule structure.

**Figure 2 ijms-24-06248-f002:**
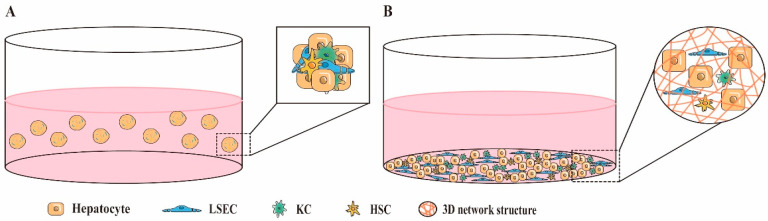
Examples of direct 3D cell co-culture liver models. (**A**) Mixed cell spheroid co-culture system. (**B**) Three-dimensional scaffold co-culture.

**Figure 3 ijms-24-06248-f003:**
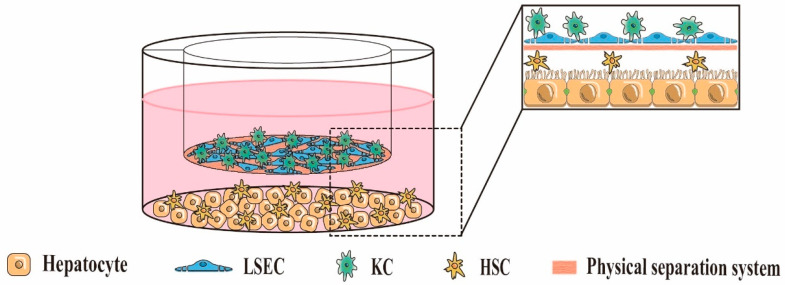
An example of an indirect 3D cell co-culture liver model.

**Table 1 ijms-24-06248-t001:** Structure and function of the major liver cell types.

Cell Type	Number (% Total Liver Cells)	Volume (% Total)	Structural Features (Physiological State/Pathological State)	Functions (Physiological State)	Reference
Hepatocytes	60%	80%	Physiological state: Rectangle; Sinusoidal surface with microvilli structure	Participate in the metabolism, synthesis, and secretion of exogenous and endogenous substances; Lipid storage; Transformation of toxic substances.	[20,22,23]
Liver sinusoidal endothelial cells (LSECs)	15–20%	3%	Physiological state: long spindle; Lacking basement membrane; Possessing sieve-like fenestrae structure through the cell Pathological state: Basement membrane formation; Fenestration structure reduced or disappeared	Regulate vascular tone; Secrete NO; Present antigen; Filter toxins and antigenic substances in portal blood.	[24,25,26,27,28,29]
Hepatic stellate cells (HSCs)	3–8%	1.5%	Physiological state: Polygon; With raised cytoplasmic pseudopods Pathological state: Myofibroblast-like phenotype	Store and release vitamin A in the body; Maintain the regeneration ability of liver tissue; Regulate the immune function of the liver; Maintain normal sinusoidal tone and liver stiffness by secreting pro-inflammatory and anti-inflammatory cytokines, as well as extracellular matrix.	[21,25,30,31]
Kupffer cells (KCs)	8–12%	2%	Physiological state: Elongate or nearly rounded; Morphological variability Pathological state: the M_1_ phenotype	Modulate liver immune response; Maintain hepatic iron, cholesterol and bilirubin metabolism; Remove pathogens, toxins, senescent red blood cells and platelets from the blood.	[4,32]

**Table 2 ijms-24-06248-t002:** Common 3D liver cell co-culture systems for direct 3D co-culture and/or indirect 3D co-culture, and their advantages and disadvantages.

3D Cell Co-Culture Method	3D Cell Co-Culture System	Introduction	Advantages	Disadvantages	Reference
Direct 3D co-culture (Physical contact and paracrine signaling interactions exist between heterotypic cells)	Self-aggregating multicellular spheroids	Liver cells self-aggregate into cell spheroids in ultra-low adhesion plates or using the suspension drop technique	Easy to operate; Low cost; High throughput	Hypoxia and necrosis of cells in the center of spheroid; Difficultly in controlling the size of the spheroid; Cells lack the support of exogenous matrix; The spheroid is loose; Uncontrollable spatial arrangement of cells; Cell separation is a tedious process	[65,66,67,68,69]
	3D liver organoids	Stem cells differentiated by multiple lineages or hepatocyte-like cells derived from them are co-cultured with non-parenchymal cells to form self-organizing 3D structures resembling natural liver structures	Complex structure and communication similar to the natural liver; Multiple liver cells can be derived simultaneously; Long-term expansion; Genetic background can be preserved; Genes can be manipulated; High throughput	Cell maturity heterogeneity; Specific reagents are required to induce cell differentiation; Low liver phenotype and functional maturity; Uncontrollable spatial arrangement of cells; Poor repeatability; High cost; Time-consuming	[70,71,72,73]
	3D scaffold co-culture of liver cells	Cells grow attached to porous 3D network scaffolds composed of natural materials (collagen, Matrigel, etc.) or synthetic materials (self-assembling peptides, polystyrene, etc.)	Controllable size and shape; Presence of in vivo-like biochemical and biomechanical microenvironment; Adjustable mechanical and degradation properties of synthetic scaffolds; Existence of cell–ECM interactions	Difficulty in cell-scaffold separation; Difficulty in live cell imaging; Hypoxia and necrosis of cells in the center of the scaffold; Uncontrollable spatial arrangement of cells; Unknown composition and batch-to-batch variation of natural scaffolds; Biocompatibility and cytotoxicity issues with synthetic scaffolds	[67,74,75,76,77]
Indirect 3D co-culture (Paracrine signaling interactions exist between cells, but no physical contact)	3D liver cell co-culture system based on Transwell chambers	3D layered co-culture of cells using Transwell chambers as a physical separation system	Mimics the layered structure of natural liver sinusoids; Easy to study intercellular paracrine interactions independently; Avoids unnecessary intercellular contact; Controllable spatial arrangement of cells; Easy separation of co-cultured cells for individual analysis; High throughput	High cost; Lack of physical contact between heterotypic cells	[23,78,79]
Direct 3D co-culture/Indirect 3D co-culture	3D bioprinting co-culture of liver cells	Using printing technologies such as Laser based-, Inkjet based- and bio-extrusion, biological materials (bio-ink) that act as extracellular matrix and living cells can be precisely located layer by layer, to form 3D tissue	High throughput; Excellent stability; Enables precise control of model structure and spatial arrangement of cells	High cost; Low resolution; Cell sedimentation during printing; Limited cell density; Scarcity of available bio-ink materials; The printing process can cause cell damage (phototoxicity and crush damage); The viscosity of bio-ink affects the printing performance; Deformation of scaffolds over time	[80,81,82,83]
	Microfluidic multicellular liver chips	Cells are grown in a microarray with the continuous medium flow and microstructural features of liver lobules	Similar physiological environment to liver lobules; Continuous culture-medium perfusion; Physiologically related oxygen and nutrient gradients; Continuous oxygen and metabolic waste delivery; Reproducing physiological shear stress; Controllable spatial arrangement of cells	Lack of physical contact between heterotypic cells; High cost; Operating complexity; High technical requirements; Low cell recovery rate	[12,66,84,85]

## Data Availability

Data sharing is not applicable.

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
