# Peer review of "Three-Dimensional Cell Co-Culture Liver Models and Their Applications in Pharmaceutical Research"

_ijms, 2023, doi:10.3390/ijms24076248_

Round 1

Reviewer 1 Report

It is clear that the liver is one of important organs considered in pharmaceutical research. In particular, the role of liver is very important in terms of drug toxicity and pharmacokinetics. For this reason, the need for an in vitro hepatic model similar to the human body has continued in the early stages of new drug development. In this manuscript, various 3D liver models were reviewed. It is expected to provide basic information and direction to researchers.

If some of the contents described below are supplemented, it is thought that more reliable information will be provided to the reader.

1. Many types of 3D cell co-culture models have been reported in previous studies. Then, are there any cases of drugs developed using the 3D lever model in the process of drug development? If there are actual application case, how about mentioning them in the “introduction” part?

It is likely to be more advantageous in order to emphasis for the need for a 3D cell co-culture model.

2. In Table 2, the author classified the microfluidic chip as one of the 3D cell culture. Recently, various microfluidic-based chips have been developed. The microfluidic chip differs from the existing static 3D culture in that it has a medium flow. If you mention your opinions on this in this manuscript, I think it will help readers a lot of information about novel types of 3D cell co-culture models, such as microfluidic chips.

Are there previous studies on the effect of microfluidic flow on liver functionality in 3D liver culture systems? What is your opinion about microfluidic flow during 3D culture?

3. Previous studies have shown that the ECM environment can also affect the lever function. It is one of the important factors in 3D culture of ECM. In this review, there seems to be relatively little information about ECM, such as the type of ECM and its effect on liver functionality.

How about adding more information about the ECM to this review?

Reviewer 2 Report

This review has well introduced the advantages/disadvantages and different applications of the 3D liver cell co-culture model for pharmaceutical research. Some minor issues can be fixed to polish this work.

1. Keywords: using an uncommon abbreviation, LSEC, will not help search this article; the full name may be used instead.

2. Line 64, the subject of this sentence is a bit problematic; please rephrase it.

3. Line 69, here the 2D models represent 2D co-culture or 2D monoculture models?  This question applies to the rest text whenever the 2D models appear; it may over-generalize this concept unless there is no difference between them.

4. Line 91, ‘transposing’ or ‘transporting’?

5. Line 208: please add the citation following the last sentence.

6. Tables 1 and 2, tables’ titles may be moved to the topside of the tables 

Reviewer 3 Report

This paper aimed to report 3D co-culture models of hepatocytes with non-parenchymal cells for pharmaceutical research. The contents are interesting for summarizing recent 3D co-culture liver models. Some specific points are detailed below.

1.     Introduction section is too long and not informative. It would be good to make this part a little concise, leaving only the essentials.

2.     Line 222-225

Effects of inflammatory factors on differentiation state of HepaRG should be more clearly described.

3.     Table 2

About indirect 3D co-culture model, what is the exact meaning of “Higher cost”. What does it mean to be expensive compared to what?

4.     Line 463-464

The expression “drug prediction results” is not clear and not appropriate, so it needs to be revised.

5.     Careful revision throughout the manuscript seems necessary. Abbreviations and full names are used interchangeably (ex> line 336, line 343, line 346, etc ; full name, LSECs, sECs was mix-used), or there are parts that have grammatical errors or vague descriptions.
